# Peer Victimization and Adolescent Mobile Social Addiction: Mediation of Social Anxiety and Gender Differences

**DOI:** 10.3390/ijerph191710978

**Published:** 2022-09-02

**Authors:** Wei Tu, Hui Jiang, Qingqi Liu

**Affiliations:** 1Mental Health Education Center for College Students, Department of Student Affairs, Hunan University of Science and Engineering, Yongzhou 425199, China; 2College of Education for the Future, Beijing Normal University, Zhuhai 519087, China

**Keywords:** peer victimization, mobile social addiction, social anxiety, gender, adolescents

## Abstract

Social media addiction has become one of the typical problem behaviors during adolescence. The present study examined the mediation of social anxiety between peer victimization and adolescent mobile social addiction and tested whether gender could moderate the direct and indirect effects of peer victimization. 649 adolescents between 12 and 19 years of age (*M*_age_ = 14.80, *SD*_age_ = 1.82) completed the anonymous survey. The results found that social anxiety was a mediator linking peer victimization to mobile social addiction. Gender could moderate the direct and indirect effects of peer victimization, and these two effects were stronger in girls than in boys. The results highlight the role of social anxiety in explaining how peer victimization was associated with adolescent mobile phone addiction and the role of gender in explaining when or for whom the direct and indirect associations between peer victimization and adolescent mobile social addiction were more potent. The findings would contribute to the intervention of mobile social addiction.

## 1. Introduction

Adolescents are a high incidence group of mobile phone addiction [1] and mobile phone addiction has become one of the typical problem behaviors during adolescence [2,3]. The negative influences of mobile phone addiction on adolescent physical and mental development have been tested by a large number of studies. Mobile phone addiction could predict adolescents’ sleep quality directly and indirectly through increased ruminative response style [4]. Mobile phone addiction significantly predicted anxiety and depression among adolescents [5]. Using the longitudinal questionnaire design, researchers demonstrated that mobile phone addiction significantly predicted adolescents’ depressive symptoms two years later [6].

Social networking is one of the core functions of mobile phones, and various social service applications have attracted countless people. Mobile social addiction has been considered a critical sub-type of mobile addiction [7,8]. Mobile social addiction refers to individuals’ strong psychological carving for mobile social applications and excessive use of various mobile social applications, which causes significant negative impacts on social functions and daily life [7]. Although many studies have analyzed the influencing factors of generic mobile phone addiction, such as family factors [9,10], school factors [8,11], and peer factors [12,13], there are relatively few studies focusing on the specific type of mobile phone addiction such as mobile social addiction.

Peer victimization is an essential influencing factor of mobile phone addiction in adolescents [14]. Since peer victimization reflects the quality of peer interaction to some extent, it may be closely correlated with adolescents’ mobile social addiction. Adolescents who suffer from peer victimization may fail to meet the relatedness needs in real life [15]. According to the compensatory Internet use theory [16,17], Internet can provide individuals with an online world that has some significant differences from the offline world, and individuals whose psychological needs cannot be met in real life can seek compensatory satisfaction in the network world. Online compensatory satisfaction may encourage individuals to indulge in the online world and eventually develop Internet addiction [18]. Cross-sectional studies have confirmed the relationship between peer victimization and Internet addiction [19] or mobile phone addiction [14]. Longitudinal research also demonstrated that peer victimization significantly predicted mobile phone addiction six months later [12]. Although studies have confirmed the effect of peer victimization on adolescent mobile phone addiction, few studies have directly tested the association between peer victimization and mobile social addiction. Based on previous theoretical and empirical evidence, the present study hypothesized that peer victimization significantly predicted adolescent mobile social addiction (Hypothesis 1).

Social anxiety may be another factor that can increase the risk of mobile social addiction. Social anxiety refers to a marked and persistent fear of being scrutinized or rejected in social situations [20,21]. According to the cognitive-behavioral model of Internet addiction [22], maladaptive cognitive factors are the key proximal factor of Internet addiction. As a specific maladaptive cognitive factor, social anxiety has been proved to have a significant impact on Internet addiction by many studies [23,24,25]. Mobile phone addiction, a typical form of Internet addiction in the mobile Internet era, has also been linked to social anxiety by many researchers. Previous research confirmed the effect of social anxiety on mobile phone addiction, whether in adolescents [26], young adults [27], or the elderly [28]. However, considering the effect size in different populations, social anxiety has the strongest predictive effect on mobile phone addiction in adolescents [28]. Since social anxiety makes individuals excessively worried about the real social situation, adolescents with social anxiety generally tend to meet relatedness needs through online social network services. Mobile phones have been the dominant access to the Internet among adolescents. Adolescents with social anxiety may thus excessively use social networking services on mobile phones to establish and maintain interpersonal relationships, resulting in mobile social addiction.

In addition, social anxiety may be influenced by peer victimization. According to social self-efficacy theory [29,30], individuals with low social self-efficacy generally believe that there are many uncertain situations in social communication, and they cannot cope with emergencies in social situations. Thus, these adolescents have strong feelings of anxiety in social situations. Negative life events such as bullying victimization are the key factors that decrease social self-efficacy among adolescents [31]. Research has found that peer victimization could negatively predict social self-efficacy and, in turn, increase anxiety risk [32]. Many studies have also revealed the direct association between peer victimization and social anxiety. For instance, peer victimization was strongly correlated with adolescent social anxiety [33]. Relational victimization could predict symptoms of social phobia one year later [34]. Since peer victimization positively predicts social anxiety and social anxiety may positively predict mobile social addiction, social anxiety may be a vital mediator linking peer victimization to mobile social addiction. Thus, the present study hypothesized that social anxiety played a mediation role between peer victimization and adolescent mobile social addiction (Hypothesis 2).

Moreover, gender may have an impact on the mediation model of social anxiety. Although there is no consistent conclusion on the gender difference in peer victimization, previous research has found that females have higher levels of social anxiety [35,36,37] and mobile phone addiction [14,38] than males. Girls scored higher than boys on the Social Phobia Inventory full scale and its three sub-scales in adolescents aged 12-16 years [35]. Females are more likely to have a social anxiety disorder and they also report greater clinical severity [36]. Female college students scored higher on mobile phone addiction than males [38]. Girls scored significantly higher on the Smartphone Addiction Scale-Short Version for Adolescents than boys [14]. Evidence also revealed that females generally have higher social anxiety and mobile phone addiction when influenced by negative factors [14,37,39]. Thus, when girls encounter peer victimization, they may have higher levels of social anxiety than boys. The direct and indirect effects of peer victimization may be more potent in girls than boys. Therefore, this study hypothesized that gender moderated the direct effect of peer victimization on adolescent mobile social addiction and the mediation effect of social anxiety, with the direct and indirect effects being stronger in girls than boys (Hypothesis 3).

To sum up, the present study aimed to answer two main questions: (a) whether social anxiety could mediate the relationship between peer victimization and mobile social addiction, and (b) whether gender could moderate the direct and indirect effects of peer victimization on mobile social addiction. The results would help answer how peer victimization is associated with mobile social addiction and for whom the effect of peer victimization are strong.

## 2. Materials and Methods

### 2.1. Participants and Procedure

This study was approved by the Institutional Review Board at the first author’s affiliation. Informed consent was obtained from the schools, participants, and guardians. We selected two schools in Central China through convenience sampling. In each school, random cluster sampling was adopted to choose one or two classes in each grade from 7th grade to 12th grade. A total of 685 adolescents were invited to participate in our anonymous survey in the classrooms. After excluding 36 adolescents who failed to complete all the questionnaires or provided invalid responses, data of 649 adolescents were included in our formal analysis. Among these participants, 315 adolescents (48.54%) were girls, and 334 were boys (51.46%). The mean age of the these adolescents was 14.48 (*SD*_age_ = 1.82).

### 2.2. Measurements

#### 2.2.1. Peer Victimization

The Chinese version of the Peer victimization Scale [14] was used. It includes four items measuring four types of peer victimization (i.e., verbal victimization, physical victimization, property victimization, and relational victimization). Sample items are “how often did your peers pick on you or say mean things to you” and “how often did your peer hit you.” Participants answered the four items on a five-point scale (0 = not once in the past month, 4 = every day). Cronbach’s α for this measure was 0.67, which is in line with previous studies [14,40]. The index of confirmatory factor analysis (CFA) showed a good fit of the scale model: *χ^2^*/*df* = 6.81, RMSEA = 0.05, CFI = 0.99, NFI = 0.98, GFI = 0.99.

#### 2.2.2. Social Anxiety

The Chinese version of the Social Anxiety subscale [41] of the Self-Consciousness Scale [42] was used. It includes six items rated on a five-point scale (0 = never, 4 = always). The sample item is “I get embarrassed very easily.” Cronbach’s α for this measure was 0.83. The index of CFA showed a good fit of the scale model: *χ^2^*/*df* = 5.65, RMSEA = 0.08, CFI = 0.98, NFI = 0.98, GFI = 0.98.

#### 2.2.3. Mobile Social Addiction

The Mobile Social Addiction subscale of the Mobile Phone Addiction Type Scale (MPATS) developed in Chinese adolescents and young adults [7] was used. It includes six items rated on a five-point scale (1 = never, 5 = always). Sample items are “Every chance I get, I open the social networking apps on my phone, even if it is just for a few glances” and “I overlook interacting with my family or friends because I spend too much time socializing on my phone”. Cronbach’s α for this measure was 0.90. The index of confirmatory factor analysis (CFA) showed a good fit of the scale model: *χ^2^*/*df* = 5.62, RMSEA = 0.08, CFI = 0.99, NFI = 0.99, GFI = 0.98.

### 2.3. Main Statistical Analyses

The descriptive statistics, the independent sample *t*-test, and the Pearson correlation analysis were conducted using SPSS 23.0 (IBM Corporation, Armonk, NY, USA). The mediation and moderated mediation model analyses were conducted using the PROCESS macro for SPSS [43]. Due to the potential impacts, age and daily mobile phone use were included as covariates in the model analyses [44,45].

## 3. Results

### 3.1. Preliminary Analysis

There were significant gender differences in the scores of peer victimization, social anxiety, and mobile social addiction (see Table 1). Specifically, girls scored high on peer victimization (*t* = 3.36, *p* < 0.01), social anxiety (*t* = 3.27, *p* < 0.01), and mobile social addiction (*t* = 3.71, *p* < 0.01) than boys. In both boy and girl groups, peer victimization, social anxiety, and mobile social addiction were significantly correlated (see Table 2).

### 3.2. Mediation Model of Social Anxiety

Table 3 presents the mediation model results generated by Model 4 of the PROCESS macro for SPSS [43]. The first regression model of mobile social addiction showed the unique effect of peer victimization, excluding the effect of the mediator (social anxiety). The last regression model of mobile social addiction showed the effect of peer victimization when the mediator (social anxiety) was included. After controlling for age and daily mobile phone use time, the direct path coefficient from peer victimization to mobile social addiction in the absence of the mediator (social anxiety) was significant (β = 0.36, *p* < 0.001). In addition, peer victimization significantly predicted social anxiety (β = 0.30, *p* < 0.001). When peer victimization and social anxiety were included together, social anxiety predicted mobile social addiction (β = 0.41, *p* < 0.001), and the effect of peer victimization on mobile social addiction was still significant (β = 0.24, *p* < 0.001). The mediating effect of social anxiety was 0.12, with 95% confidence interval being [0.09, 0.16]. The mediation effect of social anxiety accounted for 33.73% of the total effect. The mediation model of social anxiety between peer victimization and adolescent mobile social addiction is illustrated in Figure 1.

### 3.3. Moderated Mediation Model of Social Anxiety and Gender

Table 4 presents the moderated mediation model results generated by Model 59 of the PROCESS macro for SPSS [43]. After controlling for age and daily mobile phone use time, peer victimization significantly predicted social anxiety (β = 0.27, *p* < 0.001), and this effect was moderated by gender (β = 0.21, *p* < 0.01). The association between peer victimization and social anxiety was stronger for girls than for boys (see Figure 2). In addition, social anxiety significantly predicted mobile social addiction (β = 0.39, *p* < 0.001), and this effect was moderated by gender (β = 0.300, *p* < 0.001). The association between social anxiety and mobile phone addiction was stronger for girls than for boys (see Figure 3). Moreover, peer victimization significantly predicted mobile social addiction (β = 0.20, *p* < 0.001), and this effect was moderated by gender (β = 0.21, *p* < 0.05). The association between peer victimization and mobile social addiction was strong in girls (β = 0.30, *p* < 0.001), but it was not significant in boys (β = 0.09, *p* > 0.05) (see Figure 4).

The conditional effect analysis presented the direct and indirect effects in boys and girls, respectively. The direct effect of peer victimization on mobile social addiction was strong in girls (β = 0.30, *p* < 0.001) but not significant in boys (β = 0.09, *p* > 0.05). The indirect effect of peer victimization on adolescent mobile social addiction through social anxiety was strong in girls (β = 0.20, *p* < 0.001) but relatively low in boys (β = 0.04, *p* < 0.01).

## 4. Discussion

This study examined the psychological mechanisms underlying peer victimization and mobile social addiction among adolescents. The results showed that girls scored higher on peer victimization, social anxiety, and mobile social addiction than boys. Peer victimization was positively correlated with adolescent mobile social addiction. Social anxiety acted as a partial mediator. Gender not only moderated the direct association between peer victimization but also moderated the mediation effect of social anxiety.

First, this study found that girls scored higher on peer victimization, social anxiety, and mobile social addiction. Previous studies have consistently confirmed that girls had higher social anxiety [35,36,37] and social addiction than boys [46,47]. Biological factors and social roles are the important factors explaining why females have higher social anxiety [48,49] and social addiction [50,51] than males. Regarding gender differences in peer victimization, although we found that girls experienced higher peer victimization than boys, previous research indicated that boys had significantly higher overall levels of peer victimization [52]. However, in previous studies, girls generally have higher levels of relational victimization, and boys are more likely to experience physical victimization [52,53]. Different measures and different groups of participants may be the main reasons for the inconsistent research results. Future research may consider examining the gender differences in peer victimization by using the same measures in multiple groups of participants.

Second, this study found that peer victimization was positively correlated with mobile social addiction among adolescents. Hypothesis 1 was supported. Peer interactions can exert significant impacts on individual psychology and behaviors. Although many studies have explored the peer environmental factors of mobile phone addiction [10,11,12,13,14], few studies have linked peer environment to the specific type of mobile phone addiction such as mobile social addiction. Extending previous research demonstrating the effect of peer victimization on Internet addiction or mobile phone addiction [14,19], the present study found that peer victimization directly increased the risk of mobile social addiction. Negative peer interaction experiences may lead to excessive use of mobile phones for social communication among adolescents. The results indicated that researchers should simultaneously consider the nature of risk factors and the types of mobile phone addiction so as to better analyze the effect of environmental risk factors.

Third, this study found that social anxiety could act as a critical mediator. Hypothesis 2 was supported. For the effect of peer victimization on mobile social addiction, our result is in line with previous studies [33,34]. Social anxiety disorder usually begins and develops rapidly in adolescence [54]. During adolescence, the influence of peer relationships may even exceed that of the parent-child relationship. As a serious negative peer interaction experience, peer victimization undoubtedly becomes an essential factor in inducing and aggravating adolescents’ social anxiety [55]. Adolescents with social anxiety tend to avoid offline social situations because they will have high anxiety in offline social situations. For these adolescents, mobile phones can create a relatively safe online world and provide compensatory satisfaction. However, online need satisfaction may make them avoid offline activities and become overly addicted to mobile phones [18]. The mediation of social anxiety in our study also coincides with the compensatory Internet use theory [16,17], which indicates that mobile phones have become the main tool for sensitive people to seek compensatory satisfaction. The mediation model of social anxiety suggests that peer victimization not only directly but also indirectly predicts mobile social addiction. In addition, social anxiety only partially mediated the relationship between peer victimization and mobile social addiction. Other potential factors may also act as mediators linking peer victimization to mobile social addiction among adolescents.

Fourth, this study found that gender could act as a moderator. Hypothesis 3 was supported. The direct effect of peer victimization on mobile social addiction was strong in girls but insignificant in boys. The indirect effect of peer victimization on adolescent mobile social addiction through social anxiety was strong in girls but relatively low in boys. The findings indicates that girls are more likely to experience social anxiety and engage in mobile social addiction in the face of peer victimization. Existing research has demonstrated that females tend to have high social anxiety [35,36,37] and mobile phone addiction [14,38]. During adolescence, girls are more likely to encounter relational victimization [34,56]. Compared with other forms of peer victimization, relational victimization will directly influence girls’ social efficacy, causing them to feel anxious about offline social situations. They may make more online social communication through mobile phones. In addition, compared with boys, girls have a higher ruminative response style [57,58]. When encountering peer victimization, high levels of rumination will also weaken the problem-solving ability and aggravate the anxiety experience among girls [59,60]. A high ruminative response style may also increase the risk of mobile phone addiction [4]. Thus, peer victimization, especially relational victimization, would cause greater social anxiety and social media addiction.

The present study has three main limitations. First, all the measures we used were self-reported questionnaires which may influence the accuracy of the results. Future research may adopt multiple informants to collect research data. Second, this study only focused on the overall level of peer victimization but did not distinguish the impact of different types of peer victimization. Different types of peer victimization may affect mobile social addiction through different mechanisms. Future research can further analyze the effects and underlying mechanisms of different types of peer victimization. Third, this research is a cross-sectional questionnaire design which cannot confirm the causal relationship of the mediation model. Future research may consider using a longitudinal study or an experimental study.

Despite the above limitation, this study still has significant theoretical and practical implications. First, this study not only confirms the direct effect of peer victimization but also uncovers the psychological mechanism underlying peer victimization and mobile social addiction. The result shows how peer victimization is directly and indirectly associated with mobile social addiction among adolescents. Second, this study illustrates the boundary condition of the association between peer victimization and mobile social addiction by examining the gender difference. The result can tell when or for whom peer victimization has a more potent effect on mobile social addiction. Third, this study can provide useful suggestions for preventing and intervening in mobile social addiction among adolescents. For instance, parents and teachers should tell adolescents some measures that can deal with peer bullying to directly reduce the risk of negative environmental factors on mobile social addiction. Family and school may also provide some social skills training for adolescents and help them improve social self-efficacy, through which adolescents may have low social anxiety and mobile social addiction. Parents and teachers should also pay special attention to girls who suffer from peer bullying because they may have high levels of social anxiety, leading to high levels of mobile social addiction. The prevention and intervention of adolescent social anxiety and mobile social addiction should take gender differences into deep consideration.

## 5. Conclusions

Peer victimization was positively associated with mobile social addiction among adolescents. Social anxiety was a mediator explaining how peer victimization was associated with adolescent mobile social addiction. Gender was a moderator explaining when or for whom the association between peer victimization and mobile social addiction was more potent. The findings would provide useful suggestions for preventing and intervening in mobile social addiction among adolescents.

## Figures and Tables

**Figure 1 ijerph-19-10978-f001:**
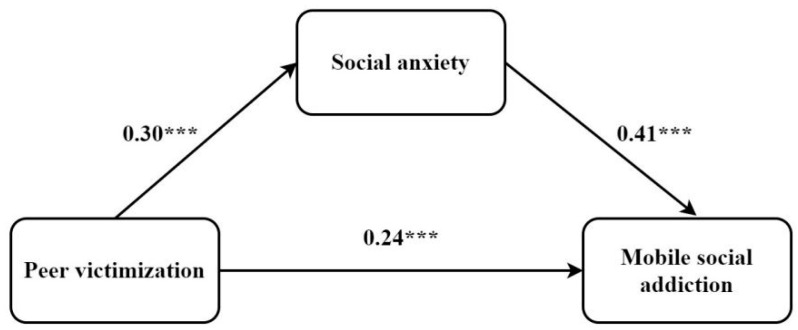
The mediation model of social anxiety between peer victimization and adolescent mobile social addiction. *** *p* < 0.001.

**Figure 2 ijerph-19-10978-f002:**
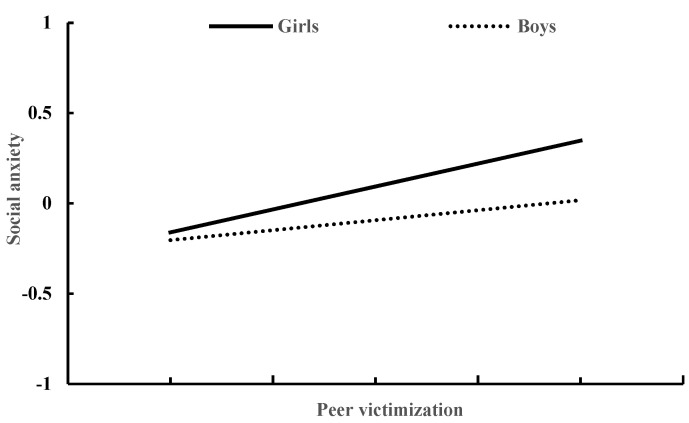
The relationship between peer victimization and social anxiety in girls and boys.

**Figure 3 ijerph-19-10978-f003:**
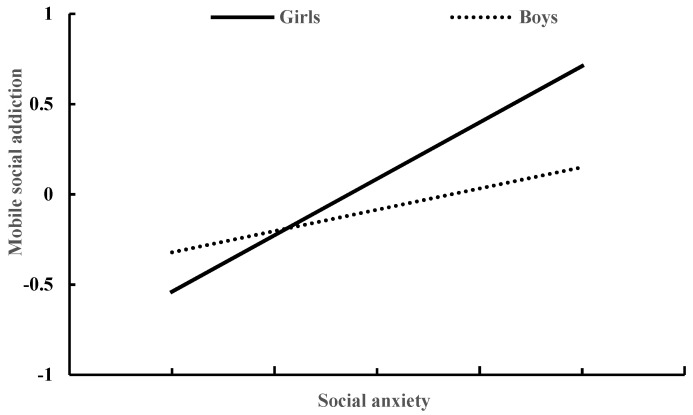
The relationship between social anxiety and mobile social addiction in girls and boys.

**Figure 4 ijerph-19-10978-f004:**
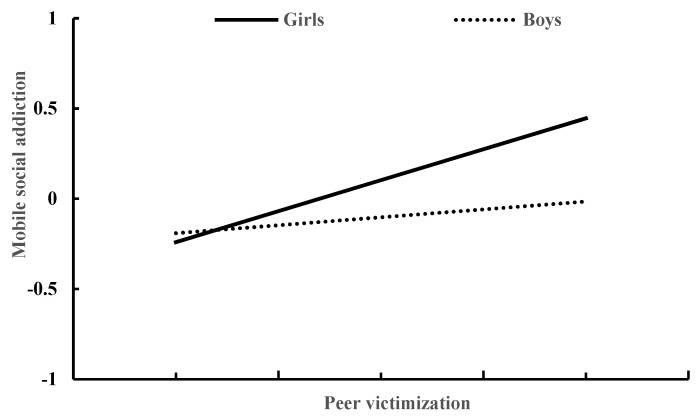
The relationship between peer victimization and mobile social addiction in girls and boys.

**Table 1 ijerph-19-10978-t001:** Test for gender differences in the variables.

Variables	Group	*M*	*SD*	*t*	*p*
Peer victimization	Girls	1.80	0.74	3.36	<0.01
	Boys	1.62	0.61		
Social anxiety	Girls	2.10	0.96	3.27	<0.01
	Boys	1.86	0.90		
Mobile social addiction	Girls	2.64	1.20	−3.71	<0.001
	Boys	2.31	1.01		

**Table 2 ijerph-19-10978-t002:** Intercorrelations between variables.

Variables	1	2	3
1. Peer victimization	-	0.16 ***	0.14 ***
2. Social anxiety	0.40 ***	-	0.28 ***
3. Mobile social addiction	0.51 ***	0.64 ***	-

Note. *** *p* < 0.001. Values above and below the diagonal represent boy and girl sample, respectively.

**Table 3 ijerph-19-10978-t003:** Mediation analysis of the role of social anxiety.

Outcome Variables	Independent Variables	β	*SE*	*t*	*p*
Mobile social addiction	Age	0.01	0.04	0.31	0.76
	Daily mobile phone use time	0.04	0.04	1.12	0.26
	Peer victimization	0.36 ***	0.04	8.53	<0.001
Social anxiety	Age	0.03	0.04	0.89	0.37
	Daily mobile phone use time	0.06 *	0.03	1.99	<0.05
	Peer victimization	0.30 ***	0.04	8.36	<0.001
Mobile social addiction	Age	−0.002	0.03	−0.07	0.95
	Daily mobile phone use time	0.02	0.04	0.43	0.67
	Peer victimization	0.24 ***	0.04	6.11	<0.001
	Social anxiety	0.41 ***	0.04	11.60	<0.001

Note. N = 649. Bootstrap sample size = 5000. LL = low limit, CI = confidence interval, UL = upper limit. * *p* < 0.05. *** *p* < 0.001.

**Table 4 ijerph-19-10978-t004:** Moderated mediation analysis.

	β	*SE*	*t*	*p*
*Mediator variable model for predicting social anxiety*				
Age	0.04	0.04	1.01	0.31
Daily mobile phone use time	0.07 *	0.03	2.13	<0.05
Gender	0.19 *	0.08	2.48	<0.05
Peer victimization	0.27 ***	0.03	7.80	<0.001
Peer victimization × Gender	0.21 **	0.07	3.18	<0.01
*Dependent variable model for predicting mobile social addiction*				
Age	0.01	0.03	0.25	0.80
Daily mobile phone use time	0.02	0.04	0.62	0.54
Gender	0.14 *	0.07	2.04	<0.05
Peer victimization	0.20 ***	0.04	4.63	<0.001
Social anxiety	0.39 ***	0.04	11.03	<0.001
Peer victimization × Gender	0.21 *	0.08	2.50	<0.05
Social anxiety × Gender	0.30 ***	0.07	4.30	<0.001
Conditional direct effect analysis at values of the moderator (gender)	*β*	*Boot SE*	*BootLLCI*	*BootULCI*
Boys	0.09	0.07	−0.05	0.24
Girls	0.30 ***	0.04	0.22	0.39
Conditional indirect effect analysis at values of the moderator (gender)	*β*	*Boot SE*	*BootLLCI*	*BootULCI*
Boys	0.04 **	0.01	0.02	0.07
Girls	0.20 ***	0.03	0.16	0.26

Note. N = 649. Bootstrap sample size = 5000. LL = low limit, CI = confidence interval, UL = upper limit. * *p* < 0.05. ** *p* < 0.01. *** *p* < 0.001.

## Data Availability

The data presented in this study are available on reasonable request from the corresponding author.

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
