# Peer review of "Peer Victimization and Adolescent Mobile Social Addiction: Mediation of Social Anxiety and Gender Differences"

_ijerph, 2022, doi:10.3390/ijerph191710978_

Round 1

Reviewer 1 Report

I thank the authors for giving me the opportunity to read this manuscript, which I find very interesting. However, I would like to make a few considerations.

Introduction:

Do not use the label for example: "Yang et al." since you are using the Vancouver bibliography.

Method:

You could better specify the method and procedure of recruitment, type of sampling, etc.

Also, you need to explain with which software the analyses were conducted (for moderation moving it from the results section).

Results:

T-test analyses are shown in the results that are not mentioned either in the method or in the explanation of the results itself.

In tab.3 better specify the difference between the first "Mobile social addiction" and the last, otherwise it looks like an error.

I recommend reporting the results of mediation models with flowcharts, with indices, so that it can be better understood.

Discussion:

I recommend reviewing how the literature references were written.

For better understanding I recommend discussing hypothesis by hypothesis. Before discussing limitations, I would try to better expand the discussions on the topic in light of the literature on the topic.

Best regards

Author Response

#Reviewer 1

I thank the authors for giving me the opportunity to read this manuscript, which I find very interesting. However, I would like to make a few considerations.

Response: Thank you very much for your constructive comments. We made several revisions to address the issues you raised.

Introduction:

Do not use the label for example: "Yang et al." since you are using the Vancouver bibliography.

Response: We deleted the labels such as “Yang et al.”

Method:

You could better specify the method and procedure of recruitment, type of sampling, etc.

Also, you need to explain with which software the analyses were conducted (for moderation moving it from the results section).

Response: Thank you very much for reminding us of this issue. We described the method, especially the procedure of recruitment and type of sampling, in detail. We also explain with which software the analyses were conducted in the Method section.  

Results:

T-test analyses are shown in the results that are not mentioned either in the method or in the explanation of the results itself.

Response: Thank you very much for highlighting this issue. We added information on T-test analyses in the Method section. We also added the explanation of the T-test results in the Discussion section.

In tab.3 better specify the difference between the first "Mobile social addiction" and the last, otherwise it looks like an error.

Response: We specified the differences between the first model of “Mobile phone addiction” and the last one.

I recommend reporting the results of mediation model with flowcharts, with indices, so that it can be better understood.

Response: Thank you very much for this suggestion. We added a figure to better report the results of the mediation model.

Discussion:

I recommend reviewing how the literature references were written.

For better understanding I recommend discussing hypothesis by hypothesis. Before discussing limitations, I would try to better expand the discussions on the topic in light of the literature on the topic.

Response: Thank you very much for highlighting this issue. We reviewed how previous studies were written. We discussed hypothesis by hypothesis. We also tried better expand the discussions on the topic in light of the literature on the topic.

Reviewer 2 Report

The authors present ‘Peer victimization and mobile social addiction in Chinese adolescents: The mediation role of social anxiety and gender differences’ including useful measures of assessment psychological conditions that is likely to be of enough interest to the managers of education program and health professionals.

 I would like to suggest several corrections or comments.

1.     Most of measures were self-reported surveys. Considering the limitations of self-reporting survey, this results should be needed objective factors, for example ‘frequency’, ‘when the peak using time’, ‘main purpose of using mobile’ etc.

2.     How could the author obtain the reliability or validity of the subjects response? Please describe it.

3.     What is the reason of excluding 36 subjects in this study, and what does it mean ‘invalid’ (3 page, 138)?

4.     Most of previous research reported gender difference of victimization, mobile addiction, social anxiety. The author should be better analysis separately by gender.

5.     Considering previous report, the boy revealed more higher peer victimization score than girls (The multidimensional peer victimization scale: A systematic review July 2018, Aggression and Violent Behavior 42)  However this study showed lower score. Pleas discuss about it.

6.     In discussion, there was not enough explanation and discussions to understand this study results for the readers. I hope that sufficient discussion and explanation of the research results have been added so that readers can understand them.

Author Response

#Reviewer 2

The authors present ‘Peer victimization and mobile social addiction in Chinese adolescents: The mediation role of social anxiety and gender differences’ including useful measures of assessment psychological conditions that is likely to be of enough interest to the managers of education program and health professionals.

I would like to suggest several corrections or comments.

Response: Thank you very much for your constructive comments. We made several revisions to address the issues you raised.

  1. Most of measures were self-reported surveys. Considering the limitations of self-reporting survey, this results should be needed objective factors, for example ‘frequency’, ‘when the peak using time’, ‘main purpose of using mobile’ etc.

Response: We agree with your opinion that self-reported surveys may impact on the accuracy of research results. We included this as a limitation in the Discussion section. In addition, we have included “daily mobile phone use time” as a covariate in the analyses to control the potential subjective impacts.

  1. How could the author obtain the reliability or validity of the subjects response? Please describe it.

Response: We reported the coefficients of the Cronbach’s α for the scales we used. The coefficients of Cronbach’s α can reflect the reliability of the responses. Regarding the validity of the responses, we conducted the confirmatory factor analysis (CFA) to test the construct validity. The indexes of CFA showed a good fit. We added the indexes of CFA in the Method section. 

  1. What is the reason of excluding 36 subjects in this study, and what does it mean ‘invalid’ (3 page, 138)?

Response: Some of these subjects failed to complete all the questionnaires. They gave very few answers to the questionnaires. Others provide invalid responses such as regular answers (1, 2, 3, 4,5, or 5, 4, 3, 2, 1, etc.)

  1. Most of previous research reported gender difference of victimization, mobile addiction, social anxiety. The author should be better analysis separately by gender.

Response: Thank you very much for highlighting this issue. The conditional direct effect analysis and the conditional indirect effect analysis showed the direct effect and indirect effect in boys and girls, respectively. We added information to highlight that the direct and indirect effects in boys and girls can be found through the results of the conditional effect analysis.  

  1. Considering previous report, the boy revealed more higher peer victimization score than girls (The multidimensional peer victimization scale: A systematic review July 2018, Aggression and Violent Behavior 42) However this study showed lower score. Pleas discuss about it.

Response: Thank you very much for this comment. We added some information to discuss why our result on the gender differences in peer victimization was inconsistent with previous research in the Discussion section.  

  1. In discussion, there was not enough explanation and discussions to understand this study results for the readers. I hope that sufficient discussion and explanation of the research results have been added so that readers can understand them.

Response: Thank you very much for reminding us of this issue. We added some statements and explanations in the Discussion section to help readers understand our results.

Round 2

Reviewer 1 Report

Ok